# Understanding Depth and Height Perception in Large Visual-Language Models

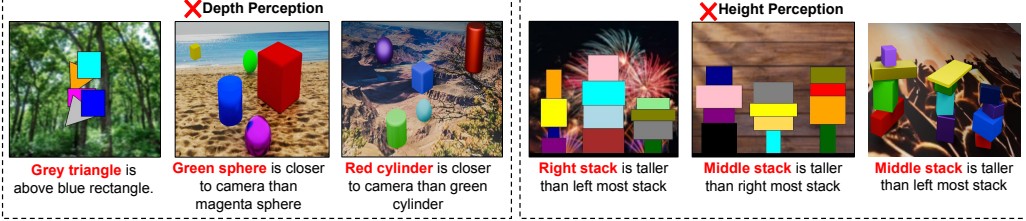

Figure 1: **Depth and height perception capability of existing VLM.** Here, we show failure cases of GPT-4V in understanding depth and height on GeoMeter, our proposed suite of benchmark datasets.

## Abstract

Geometric understanding—including depth and height perception—is fundamental to intelligence and crucial for navigating our environment. Despite the impressive capabilities of large Vision Language Models (VLMs), it remains unclear how well they possess the geometric understanding required for practical applications in visual perception. In this work, we focus on evaluating the geometric understanding of these models, specifically targeting their ability to perceive the depth and height of objects in an image. To address this, we introduce GeoMeter, a suite of benchmark datasets—encompassing 2D and 3D scenarios—to rigorously evaluate these aspects. By benchmarking 18 state-of-the-art VLMs, we found that although they excel in perceiving basic geometric properties like shape and size, they consistently struggle with depth and height perception. Our analysis reveal that these challenges stem from shortcomings in their depth and height reasoning capabilities and inherent biases. This study aims to pave the way for developing VLMs with enhanced geometric understanding by emphasizing depth and height perception as critical components necessary for real-world applications.

## 1 Introduction

In recent years, the AI community has significantly focused on integrating visual and natural language inputs, notably in Visual Question Answering (VQA) systems. These systems analyze images and answer questions about them, showing substantial advancements in understanding basic visual concepts such as shape identification (Kuhnle & Copestake, 2017), object detection (Zou et al., 2023), and the spatial relationships (Johnson et al., 2017; Chen et al., 2024; Liu et al., 2023a) by using large Visual Language Models (VLMs). These models have excelled in processing complex text and visual inputs due to their strong visual understanding capability, leading to applications in image captioning, visual question answering, image text retrieval, and so on.

The ability to understand visual properties such as size, shape, depth, and height is fundamental to visual understanding, yet many existing Visual Question Answering (VQA) benchmarks (Johnson et al., 2017; Chen et al., 2024; Liu et al., 2023a; Diwan et al., 2022; Thrush et al., 2022) do not specifically focus on the depth and height perception capabilities of Vision Language Models (VLMs). Accurate perception of these dimensions is vital for practical applications like surveillance, navigation, and assistive technologies. The lack of accurate depth and height understanding in VLMs can lead to serious consequences, such as misjudging the proximity of objects, which could result in catastrophic outcomes in real-world scenarios.

Despite VLMs' abilities to recognize object shapes and sizes, their depth and height reasoning often relies on learned size/shape cues rather than actual spatial analysis, potentially influenced by biases from training data (Jayaraman et al., 2024). Alternatively, models might estimate the depth based on the apparent size of objects, without genuine inter-object reasoning. An example illustrated in Figure 1 shows how GPT-4V (OpenAI, 2024), one of the most popular closed-source VLMs, struggles with depth perception in an image featuring two cats, despite the task being seemingly straightforward for humans. The model incorrectly assesses the spatial relationship between the cats, relying on visual cues that conflict with their actual arrangement. Additional examples in Figure 1 further demonstrate GPT-4V's failures in perceiving depth and height. These limitations highlight the need to explore such shortcomings more thoroughly and develop targeted benchmarks and training strategies that can better equip VLMs to handle complex, real-world environments with accurate depth and height perception.

In this paper, we aim to evaluate the depth and height reasoning capabilities of Vision Language Models (VLMs) to identify their strengths and limitations in visual perception. While auxiliary sensors play a crucial role in depth estimation and other alternative methods of estimating depth and height may outperform visual language models (VLMs) in specific tasks, our research aims to assess the standalone capabilities of VLMs rather than suggesting their replacement. To achieve this, we design GeoMeter, a suite of synthetic benchmark datasets focusing on 2D and 3D scenarios, named GeoMeter-2D and GeoMeter-3D respectively. These probing datasets, feature basic shapes, such as rectangles, circles, cubes, and cylinders, and are crafted to test the visual reasoning capabilities of VLMs. The development of synthetic datasets is motivated by concerns about test-time data leakage, where large VLMs, trained on vast datasets, might encounter images during testing that they have already seen during training. We prioritize clean, programmatically generated data over mere size to ensure that the evaluation is not compromised by dataset familiarity. This controlled approach minimizes the risk of data leakage and enables a more focused and precise assessment of VLMs' understanding of depth and height, free from the confounding influence of real-world cues present in many publicly sourced datasets. To this end, our probing datasets consist of around 4k unique images and 11.2k image text pairs, designed to probe depth and height reasoning in VLMs.

We extensively analyze our proposed suite of benchmark datasets on *18* recent open-source and closed-source models for the VQA task. Our findings reveal several key insights: (1) While VLMs demonstrate basic geometric understanding, they struggle significantly with depth and height perception tasks. (2) Models generally show better depth perception than height, likely due to the more common and simpler depth cues like occlusion and perspective, which are prevalent in training datasets, making depth easier to process than the more complex cues required for accurate height estimation. (3) The lack of depth and height perception ability stems from the models' intrinsic visual reasoning abilities rather than the level of prompt detail. (4) Inherent biases are evident in models' responses when faced with advanced perception tasks.

Overall, our contributions can be summarized as follows:

- We investigate the depth and height reasoning capabilities of VLMs, identifying their strengths and limitations in visual perception tasks and highlighting specific areas of improvement to enhance their visual reasoning and perception abilities.
- We conduct an extensive analysis of 18 open-source and closed-source VLMs, uncovering their behavioral patterns and inherent biases in handling depth and height perception.
- To facilitate this evaluation, we develop GeoMeter which consists of two distinct datasets: GeoMeter-2D and GeoMeter-3D, which challenge VLMs with depth and height perception tasks.

## 2 RELATED WORKS

**Visual Language Models (VLMs).** The field of AI has undergone a significant transformation with the advent of vision language models (VLMs), which are trained on extensive multimodal datasets and are versatile across numerous applications (Radford et al., 2021; Liu et al., 2023c). These models have shown remarkable performance in language and vision-related tasks, e.g. recognition, reasoning, etc. VLMs are models with a pre-trained LLM backbone and a vision encoder; which are aligned by using different methods. Recent closed-source VLMs such as GPT-4 (OpenAI, 2024), Gemini (Team et al., 2023), Claude (Anthropic, 2023) showcase a strong potential for tasks that

Table 1: **Dataset statistics** of our proposed benchmark suites. MCQ and T/F respectively denote Multiple Choice Questions and True/False questions.

| Dataset | Task | Description | Question Type | Images | Img-Text pairs |
|---------|------|-------------|---------------|--------|----------------|
| GeoMeter-2D | Depth perception | Determine which of the given objects is on the top. | MCQ, T/F | 1200 | 4800 |
| | Height perception | Provide height ordering from shortest to tallest among the given stacks | MCQ, T/F | 1200 | |
| GeoMeter-3D | Depth perception | Determine which of the given objects is closer to the camera. | MCQ, T/F | 800 | 6400 |
| | Height perception | Provide height ordering from shortest to tallest among the given stacks | MCQ, T/F | 800 | |

require understanding and processing information across different modalities. Additionally, various openly available VLMs such as LLaVA (Liu et al., 2023c), LLaVA-NeXT (Liu et al., 2023b), Bunny (He et al., 2024) etc. also have comparative performance with the closed-source models across different vision-language tasks. All of these VLMs are trained on massive amount of public and proprietary data, making them a strong performer of general reasoning.

**Exploring Visual Reasoning Capability of VLMs.** Previous works have extensively explored the spatial reasoning and object understanding capabilities of Vision Language Models (VLMs), probing their ability to grasp object-attribute relationships and spatial concepts like spatial reasoning through various benchmarks (Thrush et al., 2022; Diwan et al., 2022; Johnson et al., 2017; Krishna et al., 2017; Liu et al., 2023a; Schiappa et al., 2024; Huang et al., 2024; Tong et al., 2024). However, specific geometric properties such as depth and height perception have been largely under-explored. While there are benchmarks that assess geometric property understanding (Chen et al., 2021; Zhang et al., 2024; Sun et al., 2024), they often rely on mathematical knowledge and do not directly probe these properties in the context of natural visual understanding. Moreover, many of the datasets used in these studies (Thrush et al., 2022; Diwan et al., 2022; Krishna et al., 2017; Liu et al., 2023a; Schiappa et al., 2024; Tong et al., 2024) are curated from pre-existing datasets and/or the internet, which introduces the risk of data leakage during testing, making it difficult to assess VLMs' true capability for depth and height reasoning. Although synthetic datasets have been developed (Johnson et al., 2017; Kuhnle & Copestake, 2017); they are not specifically tailored to tasks focusing on depth and height understanding, further limiting their effectiveness in thoroughly evaluating these advanced visual concepts. Our proposed benchmark suite addresses this gap by offering image-text pairs that target depth and height perception, without relying on mathematical reasoning, providing a more focused assessment of VLMs in this area.

## 3 BENCHMARK

Our proposed suite of benchmark datasets consist of GeoMeter-2D, and GeoMeter-3D datasets that are designed to test model performance on depth and height perception tasks, utilizing unique identifiers as diverse query attributes for question generation. Table 1, and Figure 2 respectively show the dataset statistics and sample images of our proposed datasets. More samples from each dataset is given in the appendix. In the following sections, we describe the detailed data generation process for our proposed suite of benchmark datasets.

### 3.1 DATASETS

The dataset generation can be divided into two parts - Image generation (Section 3.1.1) and Question generation (Section 3.1.2).

### 3.1.1 IMAGE GENERATION

Our proposed synthetic datasets are divided into two categories - *Depth* and *Height*, with each image containing a real-world scene background to add realism while maintaining controlled, programmatically generated content. We generate images in two variety of scene density - 3 shapes and 5 shapes,

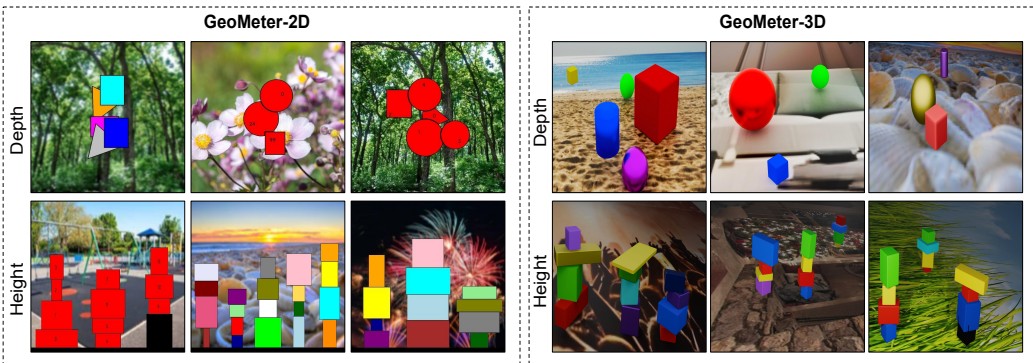

Figure 2: **Samples from the proposed suite of benchmark datasets.** Here each samples are shown with random query attributes- color and numeric label for GeoMeter-2D; and color and material for GeoMeter-3D dataset.

with each shape having one unique identifier which is used as query attribute to refer to that certain object while probing the VLMs' depth and height perception.

**GeoMeter-2D:** The GeoMeter-2D dataset includes *2400 images and 4800 unique questions*, and is designed to test depth and height perception through basic 2D shapes. The *Depth* category features overlapping 3 or 5 geometric shapes, like rectangles, triangles, and circles, positioned to create depth illusions. Ground truth for these images is stored in a scene graph that annotates each object's shape, size, color, and spatial positioning, including depth ordering through directed edges connecting overlapping objects. Each object is assigned a unique identifier based on color and numeric labels. For the *Height* category, we generated scenes featuring sequentially labeled 3 or 5 towers, each consisting of four stacked rectangles. Each tower was created by randomizing the height and width of the individual rectangles to add variability to the scene. The bottom-most rectangle in some images is placed on a black strip representing an elevated platform, making the tower effectively shorter by one rectangle in actual height but visually elevated. These images are categorized into two subgroups: *w/ step* for towers placed on a platform and *w/o step* for towers directly placed on the ground. This setup allows VLMs to be rigorously tested on height comparison tasks, requiring them to correctly interpret both the visual cues of the towers' absolute and relative heights, and the additional complexity introduced by the raised platforms. Each object in the scene is uniquely identified by its color and label, and the scene graph provides the ground truth, detailing the size, position, and elevation of each tower.

**GeoMeter-3D:** The GeoMeter-3D dataset consists of *1600 images and 6400 unique questions*, created based on the existing CLEVR dataset (Johnson et al., 2017). Scenes are generated using Blender (Community, 2018), with random jittering of light and camera positions to ensure variety. Objects in these scenes are annotated using a scene graph, which records each object's shape, size, color, material (shiny "metal" or matte "rubber"), and position on the ground plane. The *Depth* category includes randomly placed 3 or 5 cubes, spheres, and cylinders with distinct colors and materials as unique identifiers. These shapes are colored from a palette of eight colors and two materials, with increased horizontal and vertical margins than original CLEVR images between objects to reduce ambiguous spatial relationships. The scene graph captures all ground-truth information required to evaluate depth perception tasks, such as object distances and relative positions. For the *Height* category, same as the GeoMeter-2D dataset's height category setup, we created scenes with 3 or 5 towers, each consisting of four cubes stacked on top of each other. We created a base tower mesh and randomized each cube's size, color, and material (either shiny "metal" or matte "rubber") for every image. Same as GeoMeter-2D, in some scenes, the bottom-most cube is black and matte, representing an elevated platform. The ground truth for these images is represented in the scene graph, detailing the exact size, position, and elevation of each tower.

### 3.1.2 QUESTION GENERATION

The method used for generating questions is consistent across all our proposed datasets. Each question is a *Description prompt* appended with an *Answer format instruction*. The description prompt contains some general information about the scene providing semantic cues to the given image;

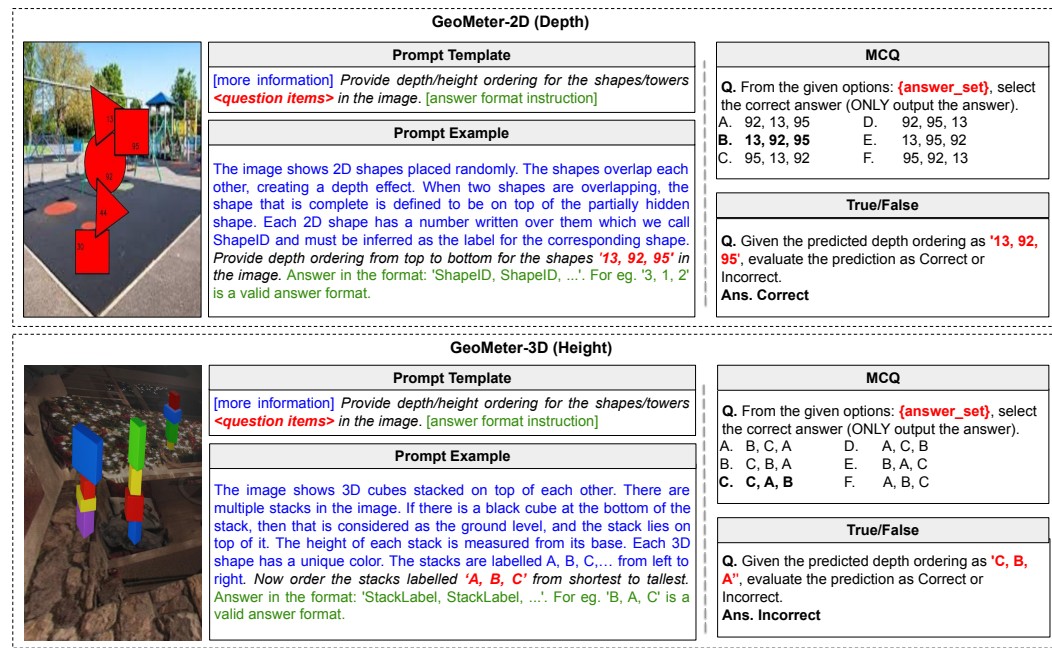

Figure 3: **Sample image-text pair from the datasets.** Here, prompt template shows the basic template for each image-text pair in our datasets, where the prompt example is the actual prompt for the image. The prompt example is appended with either MCQ or True/False type question.

followed by the actual question and answer format instruction. For example, *"[more information] Provide depth/height ordering for the shapes <question items> in the image. [more information]"* is a descriptive prompt. This is followed by *"From the given options: <answer set>, select the correct answer [more information]."* which is an answer format instruction.

The *question items* is a list containing *<query attribute>* appended by *<shape>*. Here *<query attributes>* is one of the unique identifiers of the dataset. For example in the question item *"green metal cube"*, *"green metal"* is the *<query attribute>* and *<cube>* is the shape. The *answer set* contains all possible valid values (*<query attribute>* + *<shape>*) to that given prompt. To generate both the question items and answer set, we read through the scene graph and run depth-first search on it to generate valid unambiguous values of object-pair relationship. For each image, there are two types of questions - MCQ and True/False. Some example prompts along with their corresponding image is shown in Figure 3.

## 4 EXPERIMENTAL SETUP

### 4.1 VISION LANGUAGE MODELS

We perform our benchmark evaluation on **18** state-of-the-art visual-language models. All of our chosen VLMs are trained on very large (public and/or proprietary) datasets. The selected VLMs can be categorized into **14** open-source and **4** closed-sourced models.

**Open-source models.** *LLaVA & LLaVA-NeXT* (Liu et al., 2023c;b) are a family of large open-source models combining the CLIP visual encoder (Radford et al., 2021) with the Vicuna language decoder (Chiang et al., 2023). *Fuyu-8B* (Bavishi et al., 2023) is a more efficient open-source multimodal model that projects image patches directly into the transformer, eliminating the need for an image encoder. *Bunny* (He et al., 2024) is a flexible multimodal model family offering various combinations of vision encoders and LLM backbones. *InstructBLIP* (Dai et al., 2023) leverages the BLIP-2 architecture (Li et al., 2023) for visual instruction tuning. *LLaMA-Adapter* (Gao et al., 2023) is a parameter-efficient visual instruction model, and *MiniGPT-4* (Zhu et al., 2023) aligns a frozen BLIP-2 visual encoder with the Vicuna LLM using a projection layer. We evaluate various versions of these open-source models.

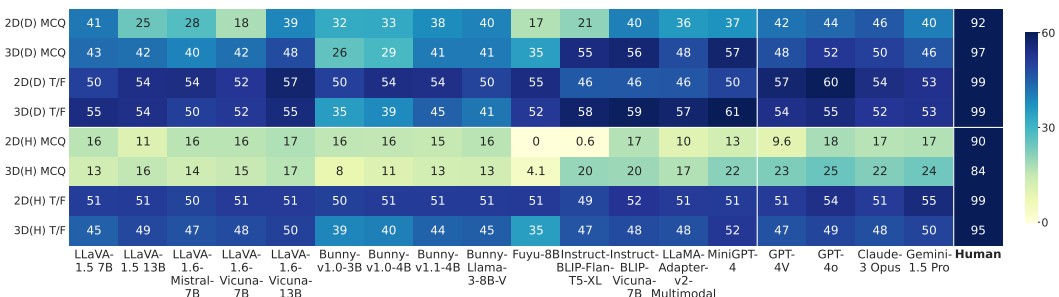

Figure 4: **Depth and height perception performance** on the proposed GeoMeter-2D and GeoMeter-3D dataset on MCQ and True/False (T/F) questions. D and H respectively denote depth, height performance. For example, 2D(D) MCQ and 2D(H) MCQ corresponds to respectively GeoMeter-2D depth and height performance on MCQ questions. Y-axis denotes the average performance across shape and query attributes and X-axis denotes all the evaluated models. Darker color denotes better performance.

**Closed-source models.** *GPT-4* (OpenAI, 2024) is a closed-source multimodal conversational model by OpenAI, based on a transformer architecture, pre-trained on large datasets and fine-tuned with Reinforcement Learning from Human Feedback (RLHF) (Christiano et al., 2017). We evaluated GPT-4V, and GPT-4o. *Claude 3 Opus* (Anthropic, 2023) is a closed-source multimodal model by Anthropic with competitive performance against other closed-source models. *Gemini 1.5 Pro* (Google, 2024) is a closed-source multimodal model by Google, surpassing GPT-4V performance across several benchmarks.

### 4.2 HUMAN EVALUATORS

We conducted a preliminary human evaluation across all of our proposed benchmark datasets, involving three evaluators who were tasked with assessing 100 uniformly sampled data from all subcategories. Similar to the model evaluation setup, each evaluator was shown one image and one prompt at a time, with a combination of multiple-choice (MCQ) and true/false questions. As illustrated in Figure 3, evaluators were asked to either select the correct depth/height ordering (MCQ) or determine whether a given prediction was accurate (T/F). The human evaluators' responses were compared against the ground truth to compute their final accuracy scores, providing a baseline for human performance on these tasks.

### 4.3 EVALUATION METRICS

We evaluate our benchmark on the task of visual question answering (VQA), with accuracy being the performance metric on MCQ and True/False type questions. Evaluation is done across query attributes and scene density for probing the VLMs' depth and height perception.

### 4.4 IMPLEMENTATION DETAILS

All models are used in accordance to the provided evaluation code and model weights. The closed-source models were accessed through APIs which have been provided through a paywall by the corresponding developing team of those models. For MCQ, the order of the given options are randomly generated, and ground truth is always randomly placed in one of those options. We have implemented already established practices (Liu et al., 2024; Suzgun et al., 2022) for creating options in multiple choice questions, randomizing both the position and the quantity of these options (up to 120 choices), and ensuring variability in the correct answer's location. For the True/False questions, the ground truth is randomly selected between True and False.

### 4.5 RESULTS

The performance of the selected models and human evaluators on the VQA task for MCQ and True/False type questions on the proposed benchmark datasets are shown in Table 2, where each model's performance represents the average accuracy of depth and height perception across all dif-

Table 2: **Performance comparison of the studied models on proposed datasets.** The reported results are averaged across depth and height category, query attributes and scene density with top scores in bold. Average denotes average performance of both datasets. Here, T/F denotes True/False type questions.

| | Model | GeoMeter-2D | | GeoMeter-3D | | Average | |
|---|---|---|---|---|---|---|---|
| | | MCQ | T/F | MCQ | T/F | MCQ | T/F |
| Open | LLaVA 1.5 7B | **28.8** | 50.5 | 28.0 | 49.8 | 28.4 | 50.2 |
| | LLaVA 1.5 13B | 17.8 | 52.5 | 29.0 | 51.3 | 23.4 | 51.9 |
| | LLaVA 1.6 Mistral 7B | 22.1 | 52.2 | 26.7 | 48.7 | 24.4 | 50.5 |
| | LLaVA 1.6 Vicuna 7B | 17.1 | 51.7 | 28.6 | 50.0 | 22.9 | 50.9 |
| | LLaVA 1.6 Vicuna 13B | 28.2 | **54.2** | 32.5 | 52.7 | 30.4 | **53.5** |
| | Bunny-v1.0-3B | 24.1 | 50.1 | 17.1 | 37.1 | 20.6 | 43.6 |
| | Bunny-v1.0-4B | 24.2 | 52.6 | 19.9 | 39.3 | 22.1 | 46.0 |
| | Bunny-v1.1-4B | 26.6 | 52.3 | 26.9 | 44.4 | 26.8 | 48.4 |
| | Bunny-Llama-3-8B-V | 27.9 | 50.2 | 26.9 | 43.2 | 27.4 | 46.7 |
| | Fuyu-8B | 8.6 | 53.0 | 19.4 | 43.2 | 14.0 | 48.1 |
| | InstructBLIP-Flan-T5-XL | 10.8 | 47.4 | 37.5 | 52.1 | 24.2 | 49.8 |
| | InstructBLIP-Vicuna-7B | 28.3 | 49.0 | 38.1 | 53.8 | **33.2** | 51.4 |
| | LLaMA-Adapter-v2-Multimodal | 22.9 | 48.8 | 32.7 | 52.4 | 27.8 | 50.6 |
| | MiniGPT-4 | 25.0 | 50.4 | **39.4** | **56.3** | 32.2 | 53.4 |
| Closed | GPT-4V | 25.5 | 54.0 | 35.2 | 50.5 | 30.4 | 52.3 |
| | GPT-4o | **30.8** | **56.7** | **38.5** | **52.4** | **34.7** | **54.6** |
| | Claude 3 Opus | 29.0 | 51.9 | 36.2 | 49.9 | 32.6 | 50.9 |
| | Gemini 1.5 Pro | 28.8 | 54.5 | 36.5 | 51.0 | 32.7 | 52.8 |
| | Human evaluators | **91.0** | **99.0** | **90.5** | **97.0** | **90.8** | **98.0** |

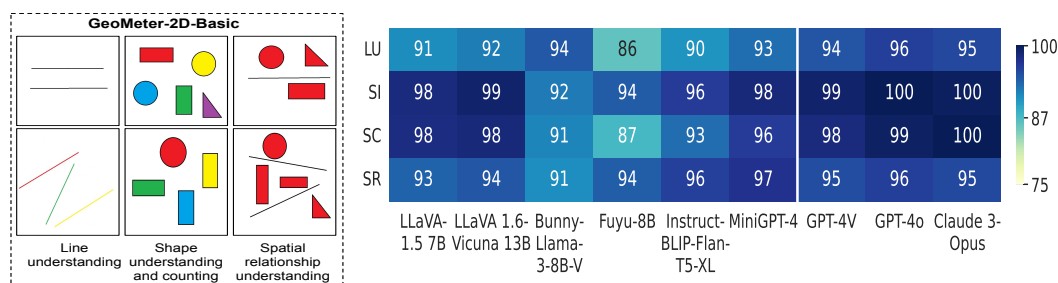

Figure 5: **Model behavior on basic understanding of shapes and size** on our created GeoMeter-2D-Basic dataset (samples on the *left*). Performance of selected models on this dataset is shown in *right*. Here, LU, SI, SC and SR respectively denote line understanding, shape identification, shape counting and spatial reasoning. Y-axis denotes performance accuracy of different categories and X-axis denotes evaluated models. Darker color denotes better performance.

ferent query attributes and scene density. Depth and height category wise results are presented in Figure 4. Additional results across all query attributes and scene density are reported in the appendix.

## 5 ANALYSIS AND DISCUSSION

### 5.1 MODEL BEHAVIOR ANALYSIS

**Human evaluations confirm tasks are straightforward.** Despite the seemingly straightforward nature of depth and height perception tasks for humans, current Vision Language Models (VLMs) struggle to achieve comparable performance. Our initial human evaluations on our datasets show consistently high accuracy in both depth and height perception tasks (Table 2, Figure 4), demonstrating that humans can effortlessly solve these tasks. In contrast, VLMs exhibit significant limitations. This performance discrepancy highlights that while these tasks may appear trivial from a human perspective, they pose substantial challenges for foundation models. Moreover, the human evaluation serves as a baseline, indicating that these tasks should be within the capability of an advanced AI system. This clear gap in model performance underscores critical limitations in VLMs' visual

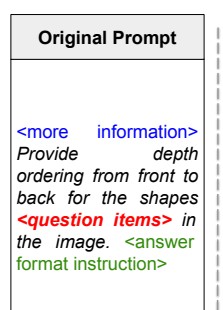

| Original Prompt | Chain of Thought Prompt |
|---|---|
| | <more information> |
| <more information> *Provide depth ordering from front to back for the shapes* **<question items>** *in the image.* <answer format instruction> | Let's think step by step.
**Step 1:** Identify the Shapes and Their Colors: Observe the image carefully and list the 3D shapes along with their colors. For example: "I see a red cube, a purple cylinder, and a yellow sphere."
**Step 2:** Determine the Depth Ordering of the Shapes: Focus on the relative positions of the shapes from the camera viewpoint. Look for visual cues such as overlapping shapes, size differences due to perspective, and shadows. For example: "The red cube is in front of the yellow sphere. The purple cylinder is behind the yellow sphere."
**Step 3:** Provide the Depth Ordering from Front to Back: Based on the observations from Step 2, arrange the shapes in order from the closest to the furthest from the camera. For example: "The depth ordering from front to back is: red cube, yellow sphere, purple cylinder."
Final Answer: Format the final answer as specified in the prompt. For example: "red cube, purple cylinder" |
| | *Provide depth ordering from front to back for the shapes* **<question items>** *in the image.* <answer format instruction> |

Figure 6: **Prompt engineering using chain of thought prompting.** Here the intermediate reasoning steps introduced in the engineered prompts of the GeoMeter-3D dataset is denoted by a dashed box.

reasoning, revealing that the models are not yet equipped to handle even elementary geometric understanding without additional sensory input.

**Models show basic visual reasoning capability but struggles in advance perception tasks.** We developed a specialized dataset called *GeoMeter-2D-Basic* containing 30 image-text pairs (some samples shown in Figure 5 *left*) to evaluate the fundamental visual reasoning capabilities of Vision Language Models (VLMs). This dataset focuses on basic geometric tasks like line understanding, shape recognition, shape counting, and assessing spatial relationships between shapes. The initial assessments using MCQs demonstrate high performance by models on these basic tasks, as detailed in Figure 5 *right*. Despite this proficiency in simple visual properties, results from Figure 4 highlight that these same models struggle significantly with depth and height perception tasks involving similar shapes. This discrepancy underscores the benchmark's value in identifying gaps in VLMs' capabilities to handle more complex spatial reasoning, beyond mere shape recognition.

**Height perception poses greater challenges than depth perception, especially in stacked object arrangements.** The superior performance of models in depth perception tasks, as compared to height perception (Figure 4 row 1,2 vs row 5,6), can be attributed to the prevalence of more common and simpler depth cues such as occlusion and perspective. These cues are widely available in many training datasets and are relatively easier for VLMs to interpret. On the other hand, we hypothesize that height estimation presents a more complex challenge as it involves analyzing the vertical placement of objects in the scene and interpreting relationships between object sizes and perspectives in a stacked arrangement. This type of height-related information is less frequent in the training data, making it harder for models to generalize effectively. To further support our hypothesis, we perform an analysis on single objects and stacks of objects for both depth and height tasks using a carefully curated subset of 100 images for each category from our GeoMeter-3D dataset. The analysis revealed that while the performance gap between depth and height for single objects is relatively narrow, there is a significant decline in performance for height tasks involving stacked objects.

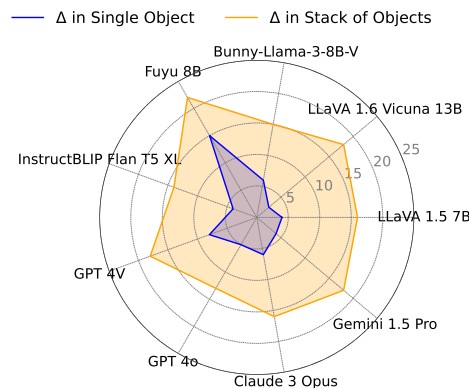

Figure 7: **Height perception is more challenging in stacked object arrangements than depth.** Here, Δ denotes performance gap between depth and height perception, which grows even larger with stacked arrangement of objects, as opposed to single objects. This suggests that while models struggle with height perception in general, stacked objects further degrade their performance.

Figure 7 shows this discrepancy of depth and height performance gap for single objects and stack of objects. This underscores our hypothesis that height perception is inherently more complex for VLMs, especially when it involves multiple objects stacked together, complicating their evaluation within a confined vertical space. Depth tasks, on the other hand, benefiting from simpler spatial cues, show better model performance.

**Models' limitation is due to inherent reasoning capability and not insufficient prompt detail.** To provide models with additional contextual information regarding visual cues with the help of intermediate reasoning, we implemented chain-of-thought prompting following (Wei et al., 2023). Chain of thought prompting enhances problem-solving by guiding models through logical reasoning steps, similar to human cognitive processes.

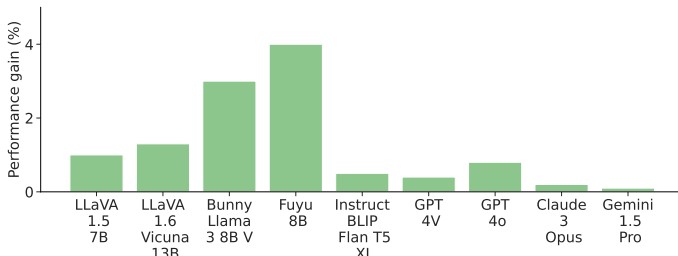

To assess its effectiveness, we selected a small subset (100 image-text pairs) of the GeoMeter-3D dataset from the depth category. We manually generated chain-of-thought prompts by rewriting the original standard prompts to include these intermediate reasoning steps, as illustrated in Figure 6. We evaluated some of the selected top-performing models using these prompts, with results shown in Figure 8. Despite

Figure 8: **Performance gain with chain of thought prompting over standard prompting** on subset of GeoMeter-3D dataset.

the highly detailed nature of these prompts, the evaluation revealed only marginal performance improvements. This suggests that even with extensive intermediate reasoning provided, the models did not benefit significantly possibly indicating that they are already performing some level of internal reasoning with the standard prompts. More importantly, this highlights that the limited performance in depth and height perception tasks is due to the inherent lower capability of the models in these areas. This is a fundamental challenge that cannot be addressed solely through prompt engineering. Instead, it points to the need for careful revisions in model architecture to improve visual reasoning capabilities in tasks involving complex spatial understanding.

**Increased scene density lowers models' perception capability.** Figure 9 shows average performance decline in the GeoMeter-2D and 3D datasets as scene density increases from 3 to 5 shapes. Open-source models like LLaVA and Bunny experience a more pronounced performance drop with increased scene complexity, while closed-source models demonstrate better resilience, suggesting they are more capable of handling visual reasoning in denser environments. However, in case of both open and closed models, the average performance drop is almost similar suggesting in general both kinds of models get affected by increased scene density.

## 5.2 MODEL BIAS ANALYSIS

We conducted further analysis on the type of prompts to study any inherent biases in the models could be influencing their performance on MCQ and True/False type questions on a smaller subset (1600 image-text pairs uniformly selected from the depth and height categories) of the GeoMeter-3D dataset.

**Some open-source models are more biased towards picking True over False than others.** The performance of some open-source models on True/False questions tends to hover around 50% (Table 2), suggesting they might not be effectively distinguishing between true and false statements, potentially defaulting to random guesses. This is highlighted by experiments showing similar outcomes (Figure 10 *left*) when ground truth is random versus always set to "True," and a significant performance decline when it is always "False," indicating a bias towards predicting "True." This bias toward "True" may arise from imbalances in training data, where models are overexposed to affirmative statements or lack sufficient counterexamples of false statements. As a result, rather than demonstrating genuine understanding, these models often rely on heuristic patterns or shortcuts. Furthermore, this behavior highlights a deeper issue: the models' inability to engage in more nuanced decision-making or reasoning under uncertainty. True/False questions, though simple in format, test models' grasp of logical consistency and factual correctness—an area where many open-source models falter. By exposing such tendencies, this evaluation method provides valuable insight into where these models need refinement, particularly in developing the capacity for more context-driven and accurate judgments.

**Some open source models are more biased towards picking the first choice in case of MCQ.** Experiments reveal that while closed-source models show consistent performance across various

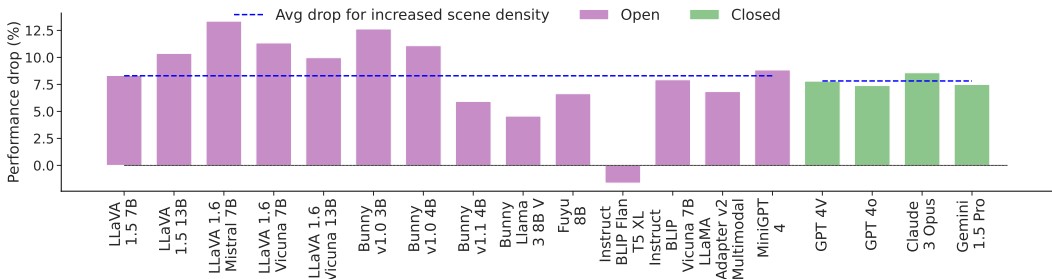

Figure 9: **Performance drop in depth perception with increased scene density.** Here, Y-axis denotes the performance drop with increased scene density (3 shapes to 5 shapes) on average of GeoMeter-2D and GeoMeter-3D datasets for MCQ. The X-axis denotes all the evaluated models

Figure 10: **Model bias analysis.** *Left:* Effect of ground truth value in True/False questions. GT-R denotes randomly set ground truth between true and false; whereas GT-T/F denotes ground truth always true or always false. *Right:* Effect of ground truth ordering in choices of MCQs. GT-C1 and GT-Ab denotes ground truth being choice 1 and not present respectively. The Y-axis denotes the average performance and X-axis denotes all the evaluated models. Darker colors denote better performance.

MCQ ground truth placements, open-source models exhibit a significant bias toward selecting the first option, particularly when the ground truth is positioned as the first choice (Figure 10 *right*). This bias could stem from the way training data is structured, where the first choice is frequently correct or if the models encounter more examples with answers listed early in the sequence, leading models to develop a preference for selecting it. Their performance drops notably when the correct answer is absent, suggesting these models struggle with identifying "None of the above" options and may rely on heuristics rather than actual reasoning, leading to random selections. This reflects a limitation in their reasoning abilities, as they likely rely on pattern recognition rather than genuine understanding of the question and its context, which suggests that open-source models may lack sophisticated decision-making processes, opting for shortcuts when faced with challenging questions.

## 6 LIMITATIONS

Our work on the depth and height perception of VLMs using synthetic datasets highlights key areas for further exploration, including the need for temporal dynamics and higher-order reasoning tasks to better understand VLM capabilities. While our benchmarks provide valuable insights, it also highlights the necessity for broader geometric reasoning and the enhancement of models' ability to process complex visual cues. Addressing these limitations will be crucial for improving VLM performance in real-world applications and extending their practical use across diverse scenarios.

## 7 CONCLUSION

Our study highlights significant challenges in the depth and height reasoning capabilities of current Vision Language Models (VLMs). While these models demonstrate basic geometric understanding and spatial reasoning, they consistently struggle with more complex visual tasks, particularly depth and height perception, which remains underdeveloped. These shortcomings are not resolved by improved prompting alone, indicating an intrinsic limitation in the models' visual reasoning abilities. Future work should focus on developing more targeted training strategies and benchmarks that address these perceptual weaknesses, particularly in height perception.

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

# 8 APPENDIX

The appendix will provide additional results on our proposed datasets. Additional results for Ge-oMeter 2D and GeoMeter 3D datasets are in Section 9.1 and Section 9.2. Sections 10, 11 respectively contain the broader impact and computational resources needed for our work.

# 9 ADDITIONAL RESULTS

## 9.1 QUANTITATIVE EVALUATION

Table 3, Table 4 present detailed results for the GeoMeter 2D dataset; and Table 5, Table 6 present detailed results for the GeoMeter 3D dataset. All of these results examine the impact of scene complexity (3 shapes vs 5 shapes), query attributes (color, labels), and question types (MCQ and True/False) on depth and height perception (respectively). While the main paper reports average results, the individual category-specific outcomes offer deeper insights. For instance, performance deteriorates with increased scene complexity (5 shapes) for many open-source models, highlighting the superior robustness of closed-source models under these conditions. Additionally, changes in query attributes show minimal impact on performance for most models, indicating their resilience to variations in query types.

Table 3: **Performance of the studied models on proposed GeoMeter-2D depth category.** Evaluation is done on the VQA task on MCQ and True/False type questions. Color, RL, PL are the query attributes. Here, RL, PL respectively denotes random numeric label, patterned numeric label.

| | Model | Depth-3 shapes | | | | | | Depth-5 shapes | | | | | |
| | | MCQ | | | T/F | | | MCQ | | | T/F | | |
| | | Color | RL | PL | Color | RL | PL | Color | RL | PL | Color | RL | PL |
|---|---|---|---|---|---|---|---|---|---|---|---|---|---|
| Open | LLaVA 1.5 7B | 48.0 | 37.5 | 54.5 | 49.0 | 54.5 | 47.0 | 36.5 | 31.0 | 39.0 | 45.0 | 56.0 | 49.5 |
| | LLaVA 1.5 13B | 36.5 | 21.0 | 29.0 | 52.0 | 57.0 | 54.0 | 35.5 | 15.0 | 11.0 | 54.5 | 53.0 | 54.0 |
| | LLaVA 1.6 Mistral 7B | 44.0 | 34.5 | 25.0 | 55.5 | 54.5 | 52.5 | 28.5 | 24.0 | 11.0 | 54.0 | 56.0 | 54.0 |
| | LLaVA 1.6 Vicuna 7B | 37.0 | 20.5 | 13.0 | 54.5 | 50.5 | 49.5 | 29.0 | 7.0 | 1.0 | 50.5 | 52.5 | 55.0 |
| | LLaVA 1.6 Vicuna 13B | 35.0 | 42.0 | 62.0 | 45.5 | 53.5 | 72.0 | 28.0 | 35.5 | 32.0 | 56.0 | 54.0 | 62.5 |
| | Bunny-v1.0-3B | 41.5 | 40.5 | 38.5 | 48.0 | 45.5 | 54.0 | 31.0 | 30.0 | 13.5 | 46.5 | 52.5 | 55.0 |
| | Bunny-v1.0-4B | 38.0 | 47.0 | 33.5 | 55.5 | 55.5 | 55.5 | 26.5 | 29.5 | 22.5 | 52.5 | 53.0 | 53.0 |
| | Bunny-v1.1-4B | 45.5 | 47.5 | 33.5 | 52.5 | 55.5 | 55.5 | 34.0 | 36.0 | 31.5 | 52.5 | 53.0 | 53.0 |
| | Bunny-Llama-3-8B-V | 34.5 | 45.0 | 46.0 | 41.0 | 45.5 | 51.5 | 27.5 | 36.5 | 48.0 | 48.5 | 53.5 | 46.0 |
| | Fuyu-8B | 33.5 | 17.0 | 4.5 | 58.5 | 55.5 | 55.5 | 30.0 | 15.5 | 3.0 | 53.5 | 53.0 | 53.0 |
| | InstructBLIP-Flan-T5-XL | 45.5 | 8.5 | 0.0 | 44.5 | 44.5 | 44.5 | 32.0 | 40.0 | 0.0 | 47.0 | 47.0 | 47.0 |
| | InstructBLIP-Vicuna-7B | 43.5 | 40.0 | 59.0 | 49.5 | 44.0 | 43.0 | 32.0 | 31.0 | 34.0 | 46.5 | 47.5 | 46.0 |
| | LLaMA-Adapter-v2-Multimodal | 41.0 | 40.0 | 39.5 | 48.5 | 45.5 | 45.5 | 31.0 | 30.0 | 33.0 | 47 | 45.5 | 45.5 |
| | MiniGPT-4 | 42.0 | 41.5 | 43.0 | 52.0 | 51.5 | 51.5 | 34.0 | 32.0 | 30.0 | 48.5 | 47.5 | 47.5 |
| Closed | GPT-4V | 45.0 | 49.0 | 41.5 | 54.5 | 57.0 | 61.5 | 38.5 | 37.0 | 40.5 | 56.0 | 58.5 | 53.0 |
| | GPT-4o | 47.5 | 44.5 | 47.0 | 55.5 | 58.5 | 70.5 | 49.5 | 36.5 | 36.0 | 62.0 | 59.0 | 52.0 |
| | Claude 3 Opus | 47.5 | 40.5 | 50 | 51.5 | 51.5 | 56.5 | 36.5 | 36.0 | 41.0 | 52.5 | 51.5 | 56.0 |

## 9.2 QUALITATIVE EXAMPLES

Figure 11 displays sample predictions from both open and closed models, highlighting their challenges with depth and height perception. The examples particularly emphasize the models' inaccuracies, especially in height perception, showcasing their limitations in spatial understanding. This figure includes predictions from the best-performing models in both the open (LLaVA 1.5 7B) and closed (GPT 4o) categories. Figures 12 and 13 present examples from the GeoMeter-2D dataset, including the specific prompts for both MCQ and True/False questions, serving as visual aids for the evaluations discussed. Similarly, Figures 14 and 15, showcase samples and corresponding prompts from the GeoMeter-3D depth and height category, respectively. These figures provide insights into the different scenarios and questions used to assess depth and height perception across various data types. Additionally, Figure 16 features image-text pairs from the GeoMeter-2D Basic dataset, highlighting the initial stages of evaluating the models' capabilities in recognizing basic properties. This collection of figures effectively illustrates the range and focus of the datasets employed to test the perceptual abilities of the models.

Table 4: **Performance of the studied models on proposed GeoMeter-2D height category.** Evaluation is done on the VQA task on MCQ and True/False type questions. Color, Label are the query attributes. Here, SP, $\overline{\text{SP}}$ respectively denote w/ step, and w/o step.

| | Model | Height-3 towers $\overline{\text{SP}}$ | | | | Height-3 towers SP | | | |
|---|---|---|---|---|---|---|---|---|---|
| | | MCQ | | T/F | | MCQ | | T/F | |
| | | Color | Label | Color | Label | Color | Label | Color | Label |
| Open | LLaVA 1.5 7B | 15.5 | 18.0 | 50.0 | 54.0 | 21.0 | 16.5 | 49.5 | 57.0 |
| | LLaVA 1.5 | 15.5 | 9.0 | 49.0 | 54.0 | 14.5 | 10.0 | 49.0 | 56.9 |
| | LLaVA 1.6 Mistral 7B | 16.0 | 17.0 | 50.5 | 55.5 | 14.0 | 15.5 | 49.5 | 53.0 |
| | LLaVA 1.6 Vicuna 7B | 14.0 | 19.0 | 49.0 | 55.0 | 18.5 | 18.0 | 50.0 | 58.0 |
| | LLaVA 1.6 Vicuna 13B | 19.0 | 19.0 | 49.5 | 54.0 | 13.5 | 20.5 | 49.5 | 57.0 |
| | Bunny-v1.0-3B | 13.5 | 17.5 | 49.0 | 51.0 | 18.5 | 20.0 | 49.0 | 57.0 |
| | Bunny-v1.0-4B | 18.0 | 16.5 | 49.0 | 54.0 | 16.0 | 12.5 | 49.0 | 57.0 |
| | Bunny-v1.1-4B | 11.0 | 18.5 | 49.0 | 54.0 | 19.0 | 15.0 | 49.0 | 57.0 |
| | Bunny-Llama-3-8B-V | 15.0 | 15.5 | 49.0 | 54.5 | 14.5 | 18.0 | 49.0 | 53.5 |
| | Fuyu-8B | 0.0 | 0.0 | 45.5 | 55.0 | 0.0 | 0.0 | 53.5 | 55.0 |
| | InstructBLIP-Flan-T5-XL | 0.5 | 0.5 | 51.0 | 46.0 | 0.0 | 0.5 | 51.0 | 43.0 |
| | InstructBLIP-Vicuna-7B | 19.0 | 16.0 | 52.0 | 54.0 | 21.0 | 20.5 | 52.5 | 57.0 |
| | LLaMA-Adapter-v2-Multimodal | 11.0 | 9.0 | 52.0 | 50.0 | 13.0 | 10.0 | 53.0 | 50.0 |
| | MiniGPT-4 | 13.0 | 12.0 | 54.0 | 52.5 | 15.0 | 14.0 | 54.0 | 51.5 |
| Closed | GPT-4V | 6.5 | 7.0 | 48.0 | 55.5 | 3.0 | 10.0 | 48.5 | 56.0 |
| | GPT-4o | 21.0 | 17.0 | 57.0 | 53.0 | 17.5 | 15.5 | 51.5 | 56.5 |
| | Claude 3 Opus | 15.0 | 13.5 | 50.5 | 51.5 | 16.0 | 18.5 | 50.0 | 56.0 |

| | Model | Height-5 towers $\overline{\text{SP}}$ | | | | Height-5 towers SP | | | |
|---|---|---|---|---|---|---|---|---|---|
| | | MCQ | | T/F | | MCQ | | T/F | |
| | | Color | Label | Color | Label | Color | Label | Color | Label |
| Open | LLaVA 1.5 7B | 14.0 | 14.0 | 46.0 | 47.0 | 14.0 | 18.5 | 51.5 | 51.0 |
| | LLaVA 1.5 13B | 12.0 | 9.0 | 52.0 | 49.0 | 8.5.0 | 8.0 | 49.0 | 48.0 |
| | LLaVA 1.6 Mistral 7B | 16.0 | 14.5 | 46.0 | 46.0 | 17.5 | 20.5 | 48.0 | 51.0 |
| | LLaVA 1.6 Vicuna 7B | 16.0 | 13.5 | 51.5 | 49.5 | 16.0 | 15.0 | 48.5 | 49.0 |
| | LLaVA 1.6 Vicuna 13B | 16.5 | 16.0 | 52.0 | 49.0 | 20.0 | 14.5 | 49.0 | 49.0 |
| | Bunny-v1.0-3B | 13.0 | 11.5 | 50.5 | 44.0 | 12.5 | 19.5 | 49.0 | 50.5 |
| | Bunny-v1.0-4B | 16.0 | 14.5 | 52.0 | 49.0 | 14.0 | 17.0 | 49.0 | 49.0 |
| | Bunny-v1.1-4B | 14.5 | 13.0 | 52.0 | 49.0 | 12.0 | 18.0 | 49.0 | 49.0 |
| | Bunny-Llama-3-8B-V | 15.0 | 15.0 | 52.0 | 47.5 | 14.5 | 21.0 | 49.0 | 49.5 |
| | Fuyu-8B | 0.0 | 0.0 | 52.5 | 51.5 | 0.0 | 0.0 | 49.0 | 46.5 |
| | InstructBLIP-Flan-T5-XL | 0.0 | 1.5 | 48.0 | 51.0 | 0.0 | 1.5 | 51.0 | 51.0 |
| | InstructBLIP-Vicuna-7B | 15.0 | 11.0 | 52.5 | 49.0 | 15.0 | 16.0 | 48.5 | 49.0 |
| | LLaMA-Adapter-v2-Multimodal | 10.5 | 8.5 | 51.0 | 52 | 9.5 | 9.0 | 50.0 | 51.5 |
| | MiniGPT-4 | 13.5 | 10.0 | 52.0 | 50.0 | 12.0 | 10.5 | 51.0 | 49.5 |
| Closed | GPT-4V | 17.5 | 12.5 | 51.5 | 50.0 | 14.0 | 6.5 | 50.0 | 49.0 |
| | GPT-4o | 18.0 | 18.5 | 59.5 | 50.0 | 19.0 | 19.0 | 51.0 | 52.0 |
| | Claude 3 Opus | 19.5 | 14.0 | 48.5 | 51.5 | 13.0 | 19.5 | 47.5 | 48.5 |

## 10 BROADER IMPACT

To our understanding, there are no negative societal impacts of our work. The goal of this work was to evaluate the depth and height perception capabilities of models that may later be used in real-world settings. This research provides insights into the depth and height perception capabilities of vision language models (VLMs), significantly impacting practical applications like autonomous driving, augmented reality, and assistive technologies. This work not only advances theoretical understanding but also opens up new possibilities for real-world applications.

## 11 COMPUTATIONAL RESOURCES

All experiments were run on an internal cluster. Each run used a single NVIDIA GPU, with memory ranging from 16GB-24GB.

Table 5: **Performance of the studied models on proposed GeoMeter-3D height category.** Evaluation is done on the VQA task on MCQ and True/False type questions. Color, ColMat are the query attributes. Here, ColMat denotes color+material

|  | Model | Depth-3 shapes | | | | Depth-5 shapes | | | |
|---|---|---|---|---|---|---|---|---|---|
|  |  | MCQ | | T/F | | MCQ | | T/F | |
|  |  | Color | ColMat | Color | ColMat | Color | ColMat | Color | ColMat |
| Open | LLaVA 1.5 7B | 49.1 | 42.5 | 59.4 | 53.8 | 43.1 | 37.5 | 55.7 | 50.4 |
|  | LLaVA 1.5 13B | 51.3 | 45.9 | 61.9 | 58.4 | 37.3 | 35.1 | 50.3 | 44.3 |
|  | LLaVA 1.6 Mistral 7B | 47.1 | 45.3 | 51.9 | 50.6 | 34.8 | 30.8 | 50.3 | 48.9 |
|  | LLaVA 1.6 Vicuna 7B | 48.8 | 47.3 | 61.9 | 58.3 | 40.2 | 32.9 | 45.9 | 40.2 |
|  | LLaVA 1.6 Vicuna 13B | 51.8 | 50.3 | 64.2 | 61.2 | 48.3 | 42.9 | 50.2 | 45.9 |
|  | Bunny-v1.0-3B | 34.8 | 29.3 | 40.2 | 35.8 | 21.9 | 18.3 | 34.8 | 29.8 |
|  | Bunny-v1.0-4B | 34.2 | 30.8 | 45.3 | 43.2 | 28.2 | 23.2 | 34.9 | 30.7 |
|  | Bunny-v1.1-4B | 45.2 | 40.3 | 44.2 | 42.9 | 40.2 | 38.3 | 48.3 | 42.9 |
|  | Bunny-Llama-3-8B-V | 44.2 | 42.1 | 45.2 | 40.8 | 40.8 | 35.9 | 40.8 | 38.3 |
|  | Fuyu-8B | 41.8 | 38.4 | 59.3 | 51.8 | 30.5 | 27.5 | 48.3 | 47.2 |
|  | InstructBLIP-Flan-T5-XL | 58.3 | 54.2 | 55.3 | 51.3 | 61.9 | 59.3 | 54.9 | 53.8 |
|  | InstructBLIP-Vicuna-7B | 57.4 | 56.3 | 56.9 | 55.4 | 60.2 | 57.3 | 59.9 | 58.6 |
|  | LLaMA-Adapter-v2-Multimodal | 52.9 | 48.3 | 47.3 | 44.2 | 59.8 | 56.8 | 57.8 | 54.7 |
|  | MiniGPT-4 | 60.3 | 56.3 | 57.8 | 54.8 | 65.3 | 62.9 | 60.3 | 54.8 |
| Closed | GPT-4V | 54.3 | 50.1 | 63.9 | 60.2 | 45.3 | 40.9 | 48.4 | 43.2 |
|  | GPT-4o | 59.9 | 52.9 | 65.9 | 60.3 | 50.3 | 44.3 | 50.3 | 44.8 |
|  | Claude 3 Opus | 56.3 | 53.9 | 57.3 | 52.3 | 47.3 | 43.2 | 51.8 | 47.4 |

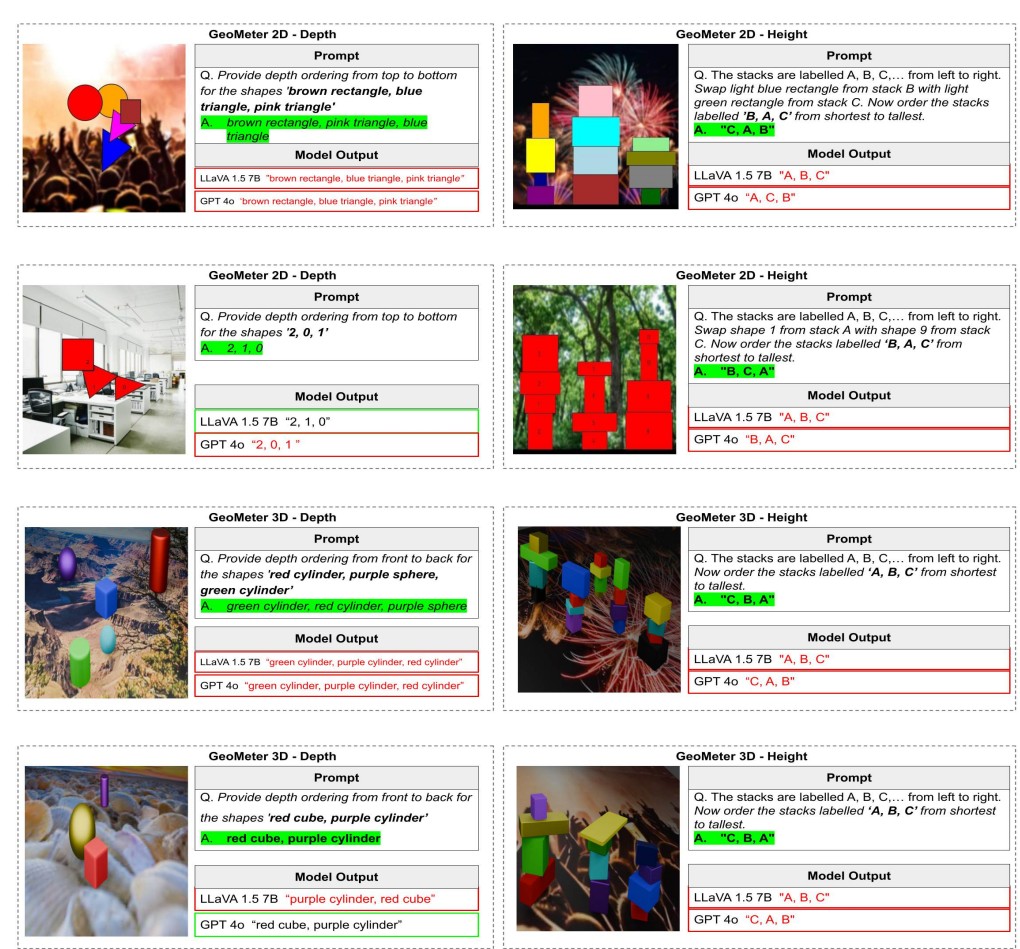

Figure 11: **Depth and height perception of open and closed models.** Here we show the prediction of LLaVA 1.5 7B and GPT 4o. Here Q and A respectively denote Question and Ground Truth Answer. Green and Red boxes respectively denote correct and incorrect prediction.

Table 6: **Performance of the studied models on proposed GeoMeter-3D height category.** Evaluation is done on the VQA task on MCQ and True/False type questions. Color, ColMat are the query attributes. Here, ColMat, SP, $\overline{\text{SP}}$ respectively denotes color+material, w/ step, and w/o step.

| | Model | Height-3 towers $\overline{\text{SP}}$ MCQ Color | ColMat | T/F Color | ColMat | Height-3 towers SP MCQ Color | ColMat | T/F Color | ColMat |
|---|---|---|---|---|---|---|---|---|---|
| Open | LLaVA 1.5 7B | 20.3 | 12.9 | 48.2 | 40.8 | 18.8 | 8.1 | 46.3 | 40.3 |
| | LLaVA 1.5 13B | 22.8 | 18.3 | 52.1 | 48.9 | 19.9 | 15.8 | 48.2 | 45.9 |
| | LLaVA 1.6 Mistral 7B | 21.9 | 18.7 | 49.9 | 42.7 | 18.3 | 12.8 | 47.9 | 44.3 |
| | LLaVA 1.6 Vicuna 7B | 20.8 | 18.9 | 48.7 | 44.8 | 18.7 | 12.7 | 49.7 | 43.8 |
| | LLaVA 1.6 Vicuna 13B | 24.9 | 19.8 | 50.7 | 47.3 | 20.8 | 17.3 | 50.2 | 45.9 |
| | Bunny-v1.0-3B | 12.4 | 9.4 | 51.4 | 50.4 | 9.4 | 5.3 | 42.9 | 40.3 |
| | Bunny-v1.0-4B | 14.9 | 10.4 | 51.8 | 48.3 | 12.9 | 10.5 | 44.3 | 41.7 |
| | Bunny-v1.1-4B | 15.9 | 12.7 | 54.8 | 52.6 | 13.7 | 11.8 | 50.3 | 48.5 |
| | Bunny-Llama-3-8B-V | 16.3 | 12.8 | 55.7 | 53.9 | 14.9 | 13.9 | 52.9 | 49.3 |
| | Fuyu-8B | 9.3 | 7.9 | 40.2 | 35.4 | 5.9 | 3.9 | 37.9 | 34.7 |
| | InstructBLIP-Flan-T5-XL | 25.1 | 20.9 | 53.8 | 50.3 | 22.9 | 20.4 | 50.3 | 48.2 |
| | InstructBLIP-Vicuna-7B | 24.9 | 21.9 | 54.3 | 52.9 | 20.8 | 18.9 | 52.7 | 49.3 |
| | LLaMA-Adapter-v2-Multimodal | 23.9 | 20.3 | 49.3 | 47.8 | 20.2 | 18.7 | 48.2 | 45.8 |
| | MiniGPT-4 | 26.9 | 24.8 | 54.8 | 53.7 | 24.8 | 20.4 | 53.8 | 51.8 |
| Closed | GPT-4V | 28.8 | 25.9 | 48.3 | 48.0 | 27.1 | 26.9 | 46.0 | 43.9 |
| | GPT-4o | 30.5 | 28.9 | 50.9 | 49.2 | 28.9 | 27.8 | 49.3 | 46.8 |
| | Claude 3 Opus | 28.3 | 24.0 | 51.8 | 48.3 | 26.1 | 22.0 | 47.3 | 43.0 |

| | Model | Height-5 towers $\overline{\text{SP}}$ MCQ Color | ColMat | T/F Color | ColMat | Height-5 towers SP MCQ Color | ColMat | T/F Color | ColMat |
|---|---|---|---|---|---|---|---|---|---|
| Open | LLaVA 1.5 7B | 12.9 | 10.4 | 48.3 | 42.3 | 10.4 | 9.3 | 47.3 | 43.8 |
| | LLaVA 1.5 13B | 13.9 | 11.3 | 50.3 | 49.2 | 11.8 | 10.5 | 49.3 | 47.3 |
| | LLaVA 1.6 Mistral 7B | 11.0 | 9.3 | 50.4 | 47.3 | 10.3 | 8.3 | 47.0 | 46.9 |
| | LLaVA 1.6 Vicuna 7B | 13.9 | 10.3 | 51.9 | 49.2 | 11.8 | 10.8 | 50.8 | 47.1 |
| | LLaVA 1.6 Vicuna 13B | 15.9 | 12.3 | 54.1 | 50.3 | 12.9 | 9.3 | 52.9 | 48.3 |
| | Bunny-v1.0-3B | 9.2 | 4.2 | 34.3 | 28.4 | 7.3 | 6.9 | 33.2 | 30.9 |
| | Bunny-v1.0-4B | 11.9 | 9.3 | 35.3 | 30.4 | 9.3 | 5.3 | 34.3 | 33.9 |
| | Bunny-v1.1-4B | 13.9 | 11.4 | 39.3 | 36.3 | 12.9 | 10.2 | 37.3 | 33.9 |
| | Bunny-Llama-3-8B-V | 13.3 | 12.1 | 38.3 | 37.9 | 10.3 | 9.9 | 36.3 | 35.9 |
| | Fuyu-8B | 4.2 | 1.8 | 35.3 | 30.0 | 0.0 | 0.0 | 32.8 | 31.9 |
| | InstructBLIP-Flan-T5-XL | 19.8 | 18.9 | 47.2 | 42.1 | 16.3 | 15.9 | 42.9 | 38.3 |
| | InstructBLIP-Vicuna-7B | 18.3 | 17.9 | 46.3 | 45.8 | 17.0 | 16.9 | 43.9 | 42.7 |
| | LLaMA-Adapter-v2-Multimodal | 15.3 | 12.8 | 48.3 | 48.0 | 13.9 | 12.8 | 47.4 | 45.4 |
| | MiniGPT-4 | 20.8 | 19.3 | 53.2 | 50.2 | 19.2 | 16.0 | 49.3 | 47.3 |
| Closed | GPT-4V | 19.3 | 17.3 | 48.4 | 47.8 | 18.3 | 16.9 | 47.0 | 46.3 |
| | GPT-4o | 22.6 | 21.9 | 51.9 | 50.3 | 20.9 | 19.6 | 49.4 | 47.4 |
| | Claude 3 Opus | 21.9 | 19.3 | 49.3 | 47.0 | 19.7 | 15.9 | 48.9 | 44.8 |

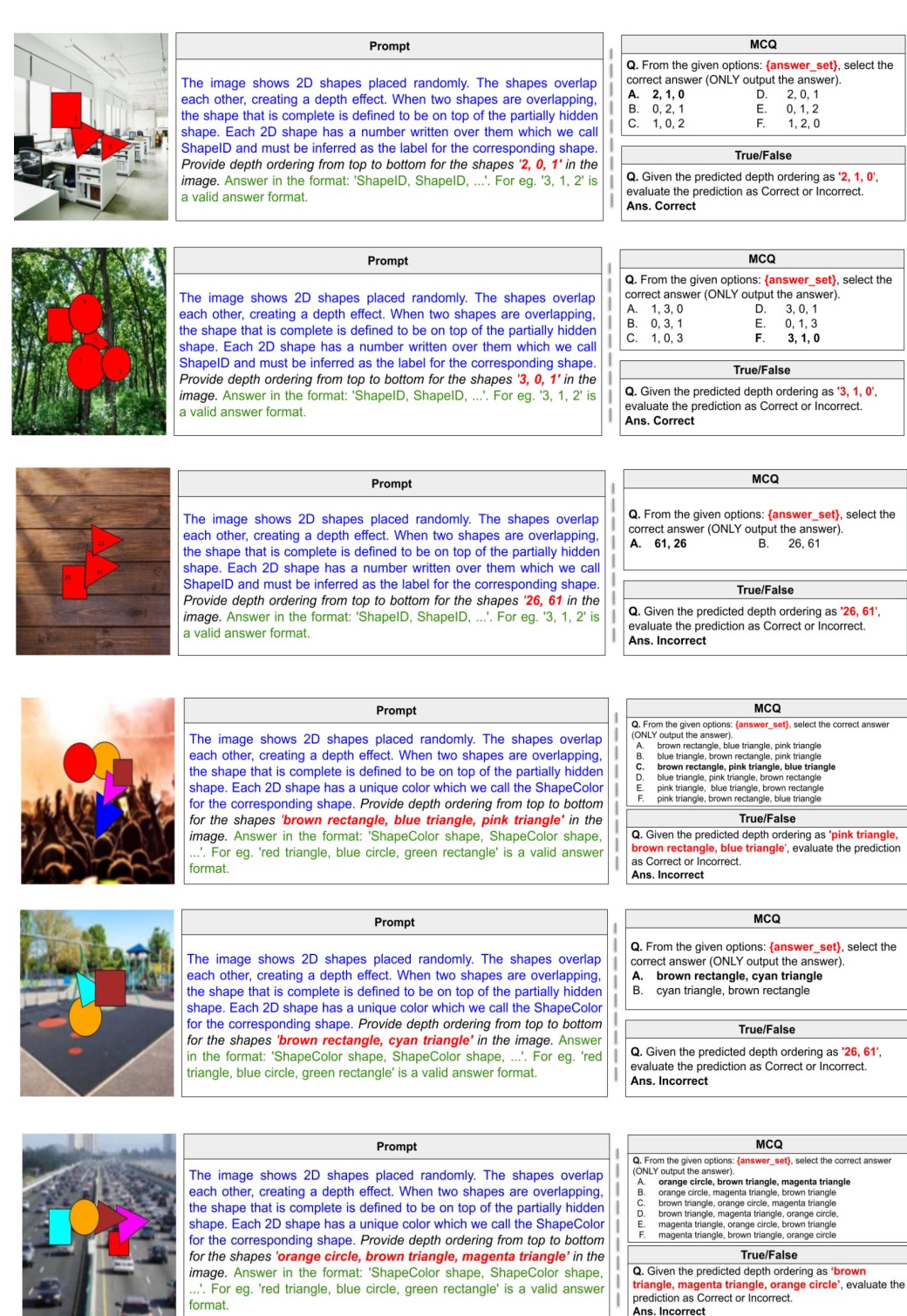

Figure 12: **Samples from GeoMeter-2D dataset - depth category.** Here each row represents one image and its corresponding prompt along with MCQ and True/False questions. First three rows show samples for labels as query attribute, whereas last three rows show samples for color as query attribute.

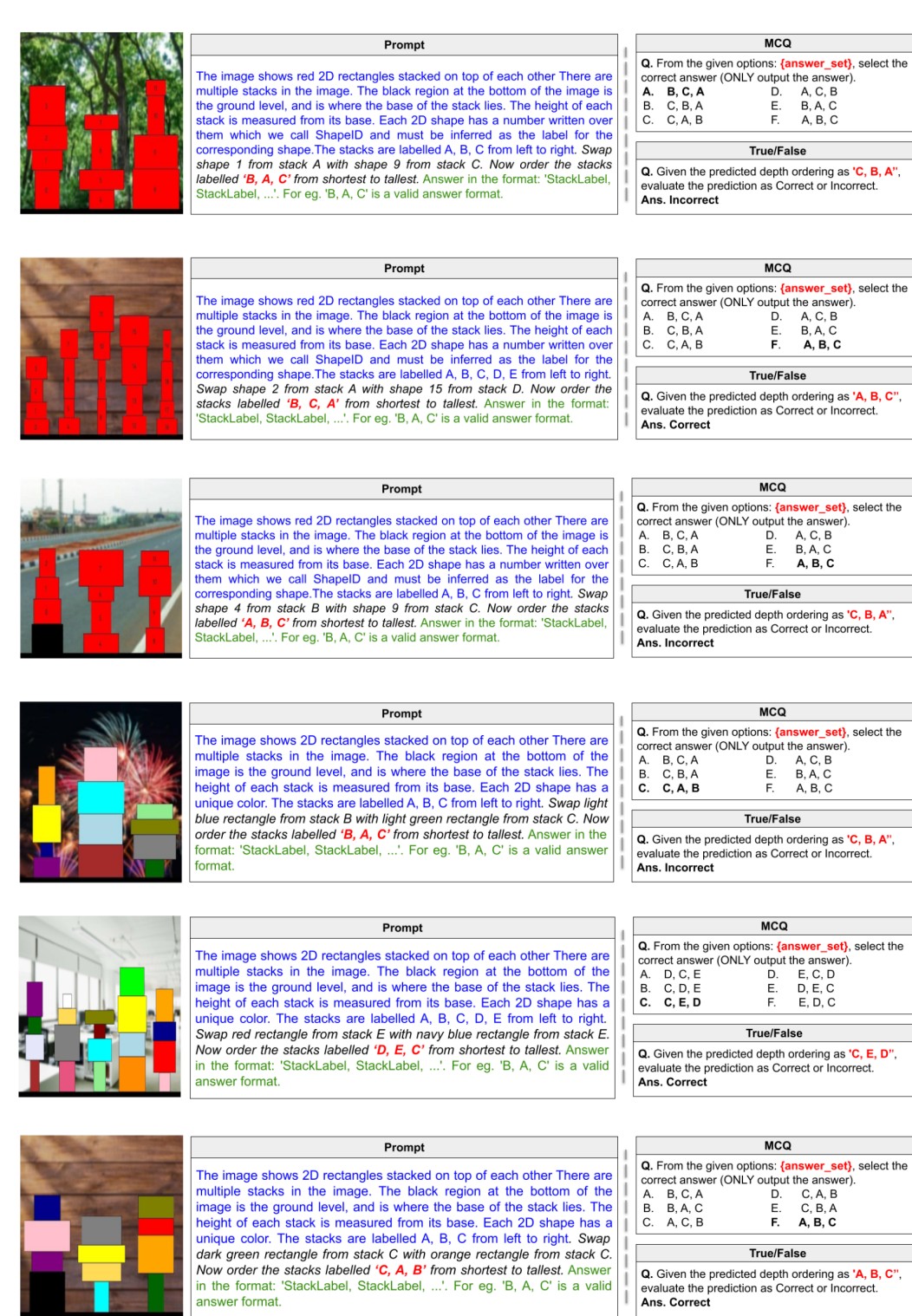

Figure 13: **Samples from GeoMeter-2D dataset - height category.** Here each row represents one image and its corresponding prompt along with MCQ and True/False questions. First three rows show samples for labels as query attribute, whereas last three rows show samples for color as query attribute

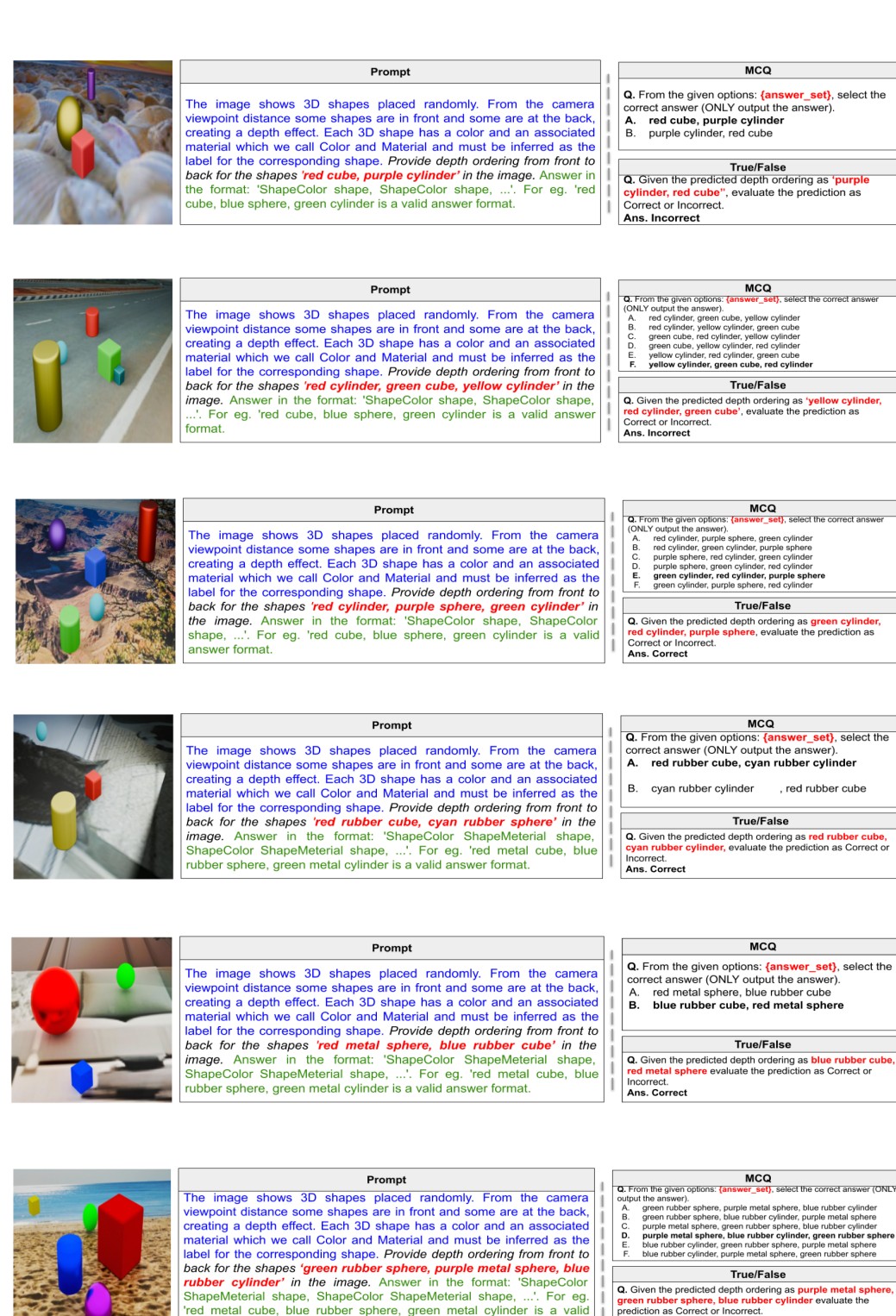

Figure 14: **Samples from GeoMeter-3D dataset - depth category.** Here each row represents one image and its corresponding prompt along with MCQ and True/False questions. First three rows show samples for color as query attribute, whereas last three rows show samples for color+material as query attribute

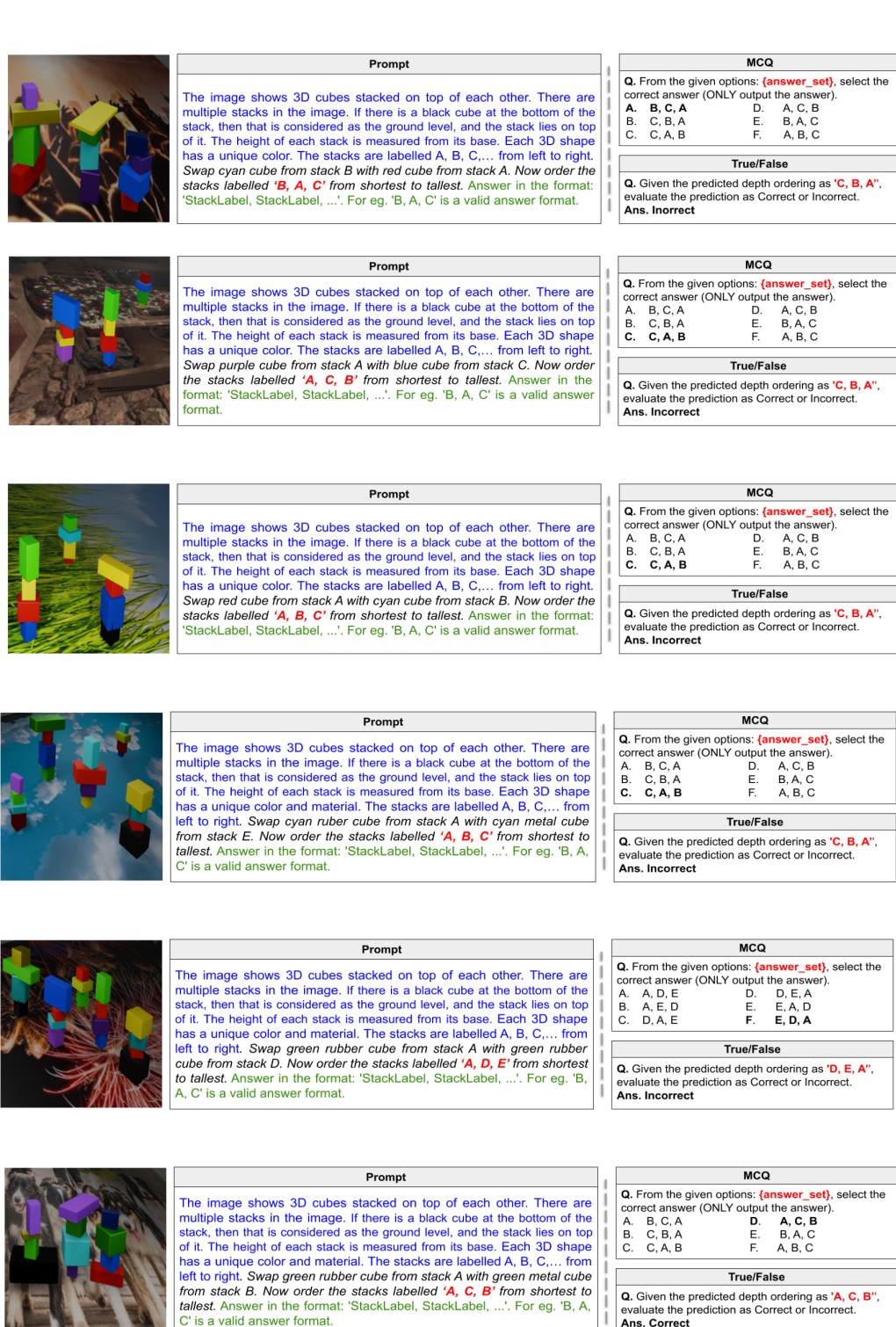

Figure 15: **Samples from GeoMeter-3D dataset - height category.** Here each row represents one image and its corresponding prompt along with MCQ and True/False questions. First three rows show samples for color as query attribute, whereas last three rows show samples for color+material as query attribute

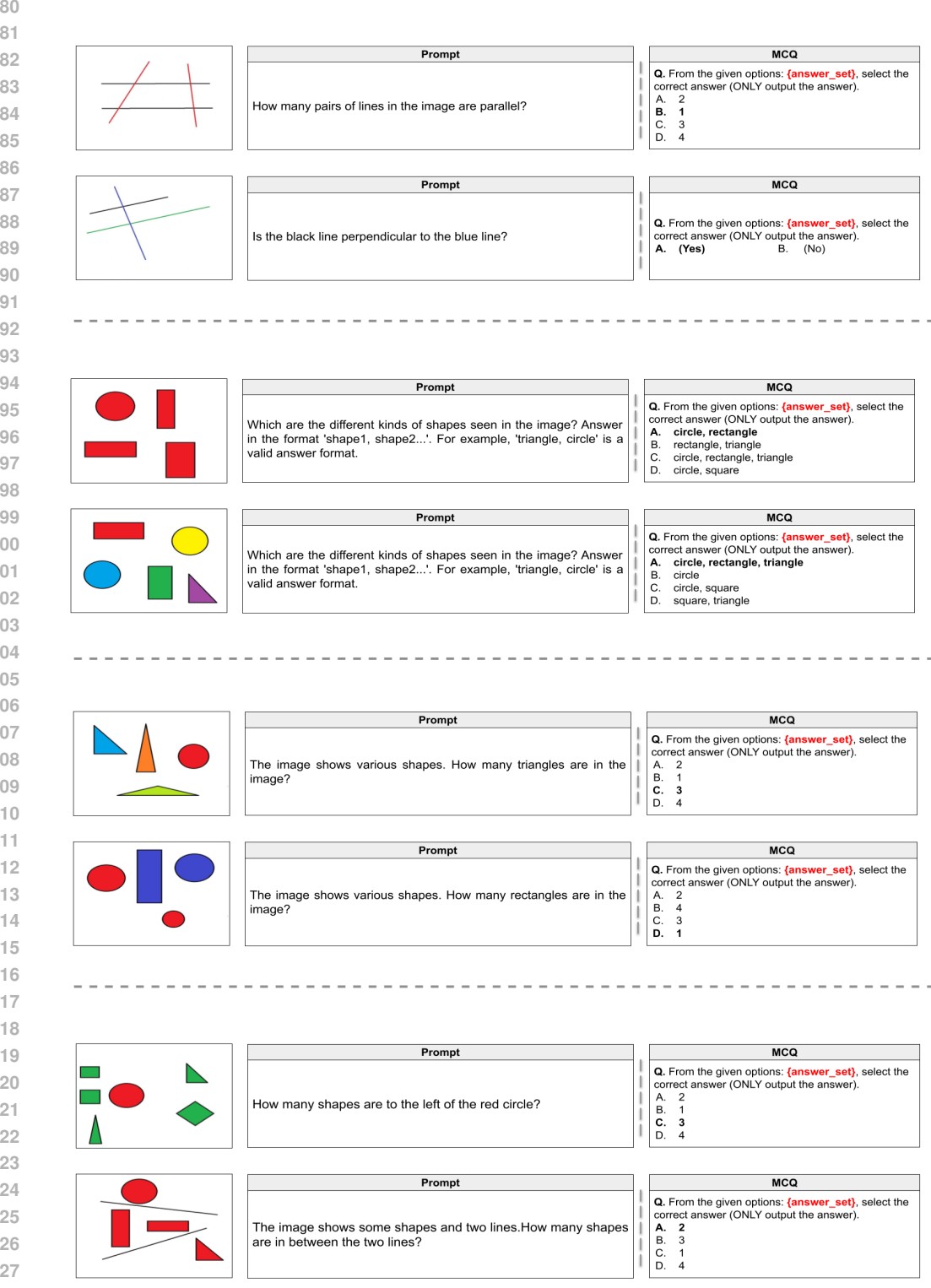

Figure 16: **Samples from GeoMeter-2D-Basic dataset.** Here each two rows respectively represent line understanding, shape identification, shape counting and spatial relationship categories. Each row shows one image and its corresponding prompt along with the MCQ.

