# OpenReview forum: "Understanding Depth and Height Perception in Large Visual-Language Models"
_ICLR.cc/2025/Conference — ICLR 2025 Conference Withdrawn Submission_

### Official Review · Reviewer_jaU8 · 2024-10-17

**Soundness:** 2
**Presentation:** 3
**Contribution:** 2
**Rating:** 5
**Confidence:** 5

**Summary:**

The paper introduces a synthetic benchmark suite for Vision Language Models (VLMs) that includes both 2D and 3D scenarios. This suite is designed to assess the current capabilities of VLMs in understanding height and depth. The evaluation and analysis are conducted on 18 different Vision Language Models.

**Strengths:**

The paper explores the spatial reasoning abilities of current Vision Language Models, an important area that currently lacks a proper benchmark in the community.

**Weaknesses:**

My main concern with this benchmark is how it was designed and whether it truly tests VLMs' ability to recognize height and depth. There are serious doubts about whether the benchmark samples properly measure these abilities or if there are flaws in the benchmark design or evaluation process. The test results, which seem to show random guessing from most models, add to these concerns. The analysis provided with the results also doesn’t offer much explanation as it fails to clarify why the models performed the way they did (even SOTA closed-source models seem random on True/False types) or give useful insights into the benchmark. Please see the details of the questions below.

**Questions:**

* It's not clear how this benchmark applies to real-world problems.
* The reason for using different materials isn't well explained, given that all objects seem to have different colors. Some samples include the material tags, while others do not (Figure 14).
* Many samples don't fit well with their backgrounds. This might make the benchmark less useful. Would a plain white background, like in CLEVR, work better?
* The numbers in GeoMeter-2D are tiny. This might be hard for VLMs (or even for humans) to recognize since VLMs often use low-res images like around 200-300 as input.
* In GeoMeter-3D (Height), the flat base often doesn't match the background image. This could confuse the model.
* It's not clear if the background matters for this test. If not, the test might be more about tricking the model than testing general height/depth recognition capabilities.
* The benchmark uses a limited range of shapes, colors, and textures. Even though there are many samples (and with diverse backgrounds), the foreground objects all look similar.
* The paper mentions that some open-source models are more biased toward picking True over False or the first option in multiple-choice questions. However, if this is true, how can these models perform well on other general VQA benchmarks?

---

### Official Review · Reviewer_cD2J · 2024-10-24

**Soundness:** 2
**Presentation:** 3
**Contribution:** 2
**Rating:** 3
**Confidence:** 4

**Summary:**

This paper proposes a new benchmark for evaluating the depth and height understanding capability of the vision language foundation models (VLMs). The benchmark has two subsets, one in 2D and another in 3D, both consisting of rendered images with synthetic objects placed on random real backgrounds. The models are asked about the depth order of the objects, or the height of a stack of rectangles. The questions are formatted into either multiple-choice or true-or-false. The authors tested the performance of 18 VLMs, including both open-source and closed-source ones, as well as human performance. All VLMs perform poorly on the benchmark.

**Strengths:**

1. The presentation of this paper is clear and easy to follow.
2. The coverage of 18 VLMs is comprehensive. Moreover, human performance is also reported, which helps understand the difficulty of the proposed questions.

**Weaknesses:**

1. Limited novelty and contribition for the depth task. The proposed 3D benchmark is highly based on existing the CLEVR dataset. CLEVR already contains questions requires reasoning about spatial relationships among objects, partially covering the depth task proposed by the authors. The CLEVR dataset has even higher complexity (up to 10 objects, compared to 5 used by the authors) and is more photo-realistic than the images shown in the paper. The authors should discuss more about the connection with CLEVR and what new aspects can the proposed dataset bring.
2. The height understanding task is novel to me, but importance of this task need justification. Depth is important for 3D understanding in human vision, but height is less straightforward, especially under the toyish setting of counting the height of a stack of building blocks. The authors should give some concrete example about how solving the proposed height task could benefit real-world applications of VLMs. Also the black platform makes the task a bit confusing to me, and the failure of VLMs may simply because they don't understand the setting.
3. The scope of the the task is limited (only 2, depth and height), and only synthetic images are used. This is a problem as more diverse and realistic benchmarks already exist. For example, a recent work BLINK[1] covers more fundamental vision tasks (including depth) with real-world images. The authors should discuss the connection with it in related work. Also since the authors use only synthetic images, it would be helpful to add discussions about the advantages or disadvantages of using synthetic images vs real-world images for benchmarking.

Overall, I feel the paper is trying to analyze an important and interesting question, i.e., the spatial relationship perception ability of VLMs. However, I feel that the current version of this paper doesn't meed the ICLR standard due to the limited scope, novelty and diversity of the proposed dataset. I would recommend the authors to cover a wider range of vision tasks in a more realistic setting. For example, covering more 3D vision tasks such as visual correspondence, object pose estimation, camera pose estimation, or using real-world images to make the benchmark more realistic.

Minor 1: L59 mentions 2 cats in Fig 1, but I see no cats in the figure. Is this a typo/mis-inserted figure?
Minor 2: From my understanding, Fig 4 and Tab 2 are basically displaying the same result, with Tab 2 simply merging the numbers for depth and height. This is kind of redundant and confusing, and I would recommend only keeping the table.

[1] Blink: Multimodal large language models can see but not perceive. ECCV 2024.

**Questions:**

N/A

**Details Of Ethics Concerns:**

1. The authors use real-world background images to render their proposed dataset but don't mention the detailed source of those images. The license and copyright of those images needs to be checked.
2. The authors use human subjects to evaluate the human performance on their dataset but didn't mention the details of the human data collection process. For example, whether a proper IRB approval is obtained.

---

### Official Review · Reviewer_JZEs · 2024-10-29

**Soundness:** 3
**Presentation:** 3
**Contribution:** 1
**Rating:** 3
**Confidence:** 4

**Summary:**

This paper mainly investigates the spatial reasoning capabilities of existing Vision Language Models (VLMs) regarding depth and height understanding. To achieve this, it proposes two synthetic benchmark datasets focusing on 2D and 3D scenarios, named GeoMeter-2D and GeoMeter-3D respectively. Each image is generated by inserting objects with random shapes and numeric labels(or materials) into a real natural image. The prompt template is designed to determine which of the given objects is on the top and provide the height ordering of the given objects to evaluate the capabilities of estimating depth and height. The authors analyze the proposed benchmark datasets on 18 recent open-source and closed-source models.

**Strengths:**

- This paper proposes two synthetic datasets to evaluate the depth and height understanding capabilities of exiting VLMs. The datasets are generated programmatically to ensure randomness.

- This paper provides a comprehensive analysis of 18 open-source and closed-source VLMs.

**Weaknesses:**

- The contribution of this paper is quite limited. The proposed benchmarks are too easy as they only contain some conceptual questions about easy spatial relationships among different objects. Some complex relationships should also be covered, like "Which object is on the left ?". And it would be great if some numerical questions could be provided as well.

- The conclusion from the analysis is not so insightful or informative. From the analysis of several VLMs, the authors find that most of them lack depth and height perception ability. I don't think this conclusion is something new to the community, there are already several papers proposed to enhance the spatial reasoning ability of the existing VLMs.

**Questions:**

I don't agree with the claim that "This study paves the way for developing VLMs with enhanced geometric understanding". Could you explain explicitly which way can be inspired by this paper?

---

### Official Review · Reviewer_pwL1 · 2024-11-04

**Soundness:** 3
**Presentation:** 3
**Contribution:** 2
**Rating:** 5
**Confidence:** 4

**Summary:**

This work proposes a benchmark to evaluate large Vision and Language Models (VLMs) on their capabilities in perceiving and understanding height and depth from single images in the format of visual question answering. They curated the dataset using synthetic shapes and symbols in both 2D and 3D and benchmarked 18 open or closed-source VLMs. Their results reveal that current VLMs still struggle with perceiving depth and height from single images, although it is easy for human evaluators.

**Strengths:**

S 1. Assessing VLMs' capabilities in height and depth perception is novel and largely not emphasized by existing VQA tasks.

S 2. Clear and detailed experiment presentation. The figures are informative. An in-depth analysis of the performances is provided.

S 3 Very reasonable ablation experiments.
  - Examined the VLMs' ability to recognize and understand shapes. It is interesting to see that all the tested models can easily understand the lines and the shapes but struggle with depth and height.
  - Chain-of-though prompting is experimented with to validate that the unsatisfactory performance cannot be addressed by simply improving the prompting techniques. This helps prove that the limitation is inherent to existing VLMs.

**Weaknesses:**

W 1.  The design choice of using simple synthetic shapes as the dataset is reasonably justified. Still, the reviewer thinks it is necessary to incorporate data sampled from the real world so that the performance on the benchmark can better indicate the performance in real life where VLMs are applied. So far this part has been missing and all data is synthetic. In other words, how consistent is the performance between this benchmark and the performance in real-life scenarios?

W 2. Similar to W 1, while the background adds realism to the data, it might be a good idea to also incorporate data with a clean and white background. The purpose is to show how much difficulty in the task is introduced by the complexity of visual input rather than depth and height. Will the performance improve if the background is clean? If so, by how much?

W 3. Add experiments on VLMs with a vision focus. While many VLMs are included in the paper, it is missing some recent ones that contributed to improving the visual perception abilities of VLMs, for example [1] and [2]. While they are cited in the paper, they are not included in the experiments. Given the content of the task, adding experiments on these VLMs (if available) seems necessary.

W 4. While exposing the perception limitations of current VLMs is a welcomed contribution, the paper will make a greater contribution to the community if it can design some new approach to try to mitigate the failures and enhance a VLM on perceiving height and depth. [3], cited in this paper, is a great example.

---
[1]  Chen, Boyuan, et al. "Spatialvlm: Endowing vision-language models with spatial reasoning capabilities." Proceedings of the IEEE/CVF Conference on Computer Vision and Pattern Recognition. 2024.

[2] Tong, Shengbang, et al. "Cambrian-1: A fully open, vision-centric exploration of multimodal llms." arXiv preprint arXiv:2406.16860 (2024).

[3] Tong, Shengbang, et al. "Eyes wide shut? exploring the visual shortcomings of multimodal llms." Proceedings of the IEEE/CVF Conference on Computer Vision and Pattern Recognition. 2024.

**Questions:**

See the weakness above.

---

### Note · Authors · 2024-11-12

I have read and agree with the venue's withdrawal policy on behalf of myself and my co-authors.